Ginsenosides and gut microbiota: differential effects on healthy individuals and irritable bowel syndrome subtypes

Du Zhi 1 2
Zhao Chengman 3
Li Jiabin 1 2
Shen Yan 2
Ren Guofei 4
Ding Jieying 1 2
Peng Jing 1 2
Ye Xiaoli 5
Miao Jing miaojing@zju.edu.cn 1 2
1 Department of Pharmacy, Children’s Hospital, Zhejiang University School of Medicine, National Clinical Research Center for Child Health , Hangzhou , Zhejiang , China
2 Research Center for Clinical Pharmacy, College of Pharmaceutical Sciences, Zhejiang University , Hangzhou , Zhejiang , China
3 Department of Gastroenterology, Affiliated Xiaoshan Hospital, Hangzhou Normal University , Hangzhou , Zhejiang , China
4 Department of Pharmacy, Affiliated Xiaoshan Hospital, Hangzhou Normal University , Hangzhou , Zhejiang , China
5 Department of Medical Administration, Children’s Hospital, Zhejiang University School of Medicine, National Clinical Research Center for Child Health , Hangzhou , Zhejiang , China
Connor Mark
Electronic publication date: 2025 Apr 15
Publication date: 2025
Volume: 13
Electronic Location ID: e19223
Received 2024 Oct 22; Accepted 2025 Mar 6
Copyright: ©2025 Du et al.
Copyright year: 2025
Copyright holder: Du et al.
License: This is an open access article distributed under the terms of the Creative Commons Attribution License, which permits unrestricted use, distribution, reproduction and adaptation in any medium and for any purpose provided that it is properly attributed. For attribution, the original author(s), title, publication source (PeerJ) and either DOI or URL of the article must be cited.
License URL: https://creativecommons.org/licenses/by/4.0/

Keywords: Ginsenosides, Gut microbiota, 16S rRNA, In vitro fermentation, Irritable bowel syndrome

Funding: Zhejiang Provincial Natural Science Foundation of China LYY22H280002 LZ23H030002 The State Administration of Traditional Chinese Medicine and the Zhejiang Province jointly build Science and Technology Program GZY-ZJ-KJ-23030 The Key Program of the Independent Design Project of National Clinical Research Center for Child Health I23I0002 This research was supported by Zhejiang Provincial Natural Science Foundation of China under Grant No. LYY22H280002 and LZ23H030002; the State Administration of Traditional Chinese Medicine and the Zhejiang Province jointly build Science and Technology Program under Grant No. GZY-ZJ-KJ-23030; the Key Program of the Independent Design Project of National Clinical Research Center for Child Health under Grant No. I23I0002. The funders had no role in study design, data collection and analysis, decision to publish, or preparation of the manuscript.

==============================
Background

Irritable bowel syndrome (IBS) is a common gastrointestinal disorder with poorly understood mechanisms. Variations in gut microbiota composition are observed in different IBS subtypes. Ginsenosides have shown potential in alleviating IBS symptoms, but their interactions with gut microbiota in different IBS subtypes are not well studied.

Methods

In this study, we investigated the effects of ginsenosides on the gut microbiota of both healthy participants and participants suffering from IBS characterized by diarrhea (IBS-D) or constipation (IBS-C), using in vitro fermentation alongside 16S rRNA sequencing and bioinformatics analyses.

Results

The analysis demonstrated that there were no statistically significant alterations in α- or β-diversity between the ginsenosides-treated and control groups across all models. However, the microbial composition assessment revealed the presence of 51 shared genera, with notable variations in composition and a significant enrichment of specific taxa. Specifically, the LEfSe analysis revealed that, following ginsenosides treatment, the healthy model groups exhibited significant enrichment of Stenotrophomonas and Achromobacter, while the IBS-D model groups demonstrated significant enrichment of Pseudomonas and Stenotrophomonas.

Conclusions

The results elucidate the distinctive microbial signatures associated with ginsenosides treatment across both healthy and IBS-D groups, underscoring the potential therapeutic efficacy of ginsenosides in modulating gut microbiota. This study highlights the necessity for further investigation into targeted microbiome therapies for IBS, which may facilitate the development of more personalized and efficacious treatment strategies for gastrointestinal health.

Introduction

Irritable bowel syndrome (IBS) is a complex gastrointestinal disorder characterized by recurrent abdominal pain or discomfort and changes in bowel habits, including diarrhea, constipation, or a combination of both (Lacy et al., 2021). IBS is divided into several specific subtypes, each defined by the primary abnormalities in bowel habits. These subtypes include IBS with diarrhea (IBS-D), which is characterized by frequent loose stools; IBS with constipation (IBS-C), which is characterized by infrequent and hard stools; mixed IBS (IBS-M), which has features of both diarrhea and constipation; and unclassified IBS (IBS-U), in which symptoms do not fit neatly into the other categories (Farmer, Wood & Ruffle, 2020). Patients with IBS often have distinct differences in their gut microbiota compared to healthy individuals. Differences in microbial composition can affect drug metabolism pathways, leading to variability in drug efficacy, safety, and inter-individual variability (Ermakov, Granados & Nigam, 2023; Ng et al., 2023).

IBS has been linked to gut microbial imbalance, thus prompting the investigation of therapeutic interventions that target and modulate these microbial communities. Probiotics, such as Bifidobacterium and Lactobacillus, have demonstrated efficacy in reducing symptoms like abdominal pain and bloating by modulating inflammation and enhancing the integrity of the intestinal barrier (Wang et al., 2020). Prebiotics, such as fructooligosaccharides, selectively promote the growth of beneficial gut microbiota. However, they may exacerbate symptoms in IBS-C and IBS-M subtypes (Wilson et al., 2019). Fecal microbiota transplantation (FMT) holds potential, though early studies report mixed outcomes, and concerns about safety and variability in donor material remain (Karimi et al., 2024). The field is seeing the emergence of treatments such as postbiotics (e.g., short-chain fatty acids) and bacteriophage therapy, which target pathogenic microbes (Sawa, Moriyama & Kinoshita, 2024; Zhao et al., 2024). Given the heterogeneity of IBS, the development of personalized approaches and the implementation of biomarker-driven trials are imperative for the improvement of treatment strategies.

Ginsenosides, bioactive compounds present in Panax ginseng, offer a promising approach to alleviate symptoms associated with IBS (Kim et al., 2005; Liu et al., 2023). In IBS-C, ginsenosides may increase gastrointestinal motility, alleviate abdominal discomfort, and improve stool consistency, while in IBS-D, they regulate intestinal transit time, reduce stool frequency, and mitigate diarrhea episodes (Chen et al., 2014; Sinagra et al., 2017). However, the systemic absorption of ginsenosides is limited, their presence in the gastrointestinal tract facilitates direct interaction with the gut microbiota (Yang et al., 2020).

In vitro fermentation of the fecal gut microbiota is a valuable approach to exploring the interplay between gut health and specific compounds, encompassing foods and drugs (Zhao et al., 2023b). This fermentation model functions as a closed anaerobic system, facilitating the study of how different compounds affect the human gut microbiota (Pi et al., 2024). Consequently, the use of individual fecal samples for in vitro fermentation offers an efficient and practical method for assessing fermentation dynamics and observing short-term shifts in microbiota composition in response to specific compounds.

This study seeks to clarify the differential effects of ginsenosides on gut microbiota through the application of in vitro fermentation techniques, focusing on both healthy individuals and patients exhibiting various subtypes of IBS. By investigating these interactions, the research aims to enhance our understanding of how ginsenosides influence gut microbiota dynamics. This knowledge may provide valuable insights into the potential therapeutic roles of ginsenosides in modulating gut health, particularly in individuals with IBS.

Materials & Methods

Drugs and reagents

The ginsenosides were purchased from Chengdu Biopurify Phytochemicals Co., Ltd., China. The ginsenosides mainly include (w/w) ginsenoside Rg1 (11.3%), Rd (17.27%), Re (23.5%), Rb1 (1.23%), Rb2 (4.3%), Rf (0.01%), and Rc (2.55%). Yeast extract, tryptone, L-cysteine, bile salts, and heme were procured from Sigma Company (St. Louis, MO, USA). MgSO4, NaCl, CaCl2, pKH2PO4, K2HPO4, metaphosphoric acid, phosphate-buffered saline (PBS), and crotonic acid were sourced from Sangon Biotech Co., Ltd. (Shanghai, China).

Participants

The study included 18 participants, aged 20 to 60 years, who were selected based on the Rome IV criteria. Among them, six participants were diagnosed with IBS-C, six with IBS-D, and six were healthy controls. Exclusion criteria included: (1) experiencing abdominal pain or discomfort for less than two days during the screening week, (2) following a highly restrictive diet (e.g., vegan), (3) recently undergoing bowel preparation for diagnostic procedures, or (4) using antibiotics, prebiotics, or probiotics within the past four weeks. Detailed inclusion and exclusion criteria can be found in the source (Staudacher et al., 2021). Participants were recruited at Xiaoshan Hospital in Hangzhou, China, between January 2022 and November 2023. Prior to their participation in the study, all participants were provided with written information regarding the study and signed a written informed consent form.

Collection of fecal samples

Participants were instructed to collect fresh fecal samples using sterile 30 mL fecal collection containers (91 mm × 24 mm; BioRise Co., Ltd., China). A minimum of four g of partially processed fecal matter, containing minimal undigested food residues and limited exposure to air after defecation, was transferred to an anaerobic jar and subsequently stored at 4 °C. All samples were processed and analyzed within four hours of collection. The study protocol was reviewed and approved by the Ethics Committee of Hangzhou Centers for Disease Control and Prevention (Approval No. 202047).

Extraction of gut microbiota

Fecal samples were stored in three sterile 1.5 mL centrifuge tubes, each containing 0.2 g of fresh fecal matter, were stored at −80 °C as backup samples. Simultaneously, 0.8 g of fresh feces was mixed with eight mL of sterile PBS in 10 mL sterile centrifuge tubes, sealed, and thoroughly shaken for uniform mixing. Following filtration, the supernatant was collected to produce a 10% gut microbiota extraction solution.

In vitro fermentation

The extracted gut microbiota was introduced into a simulated gut fermentation system and incubated at 37 °C for 24 h under strict anaerobic and airtight conditions, following the procedure outlined in previous studies (Liu et al., 2020). The composition of the Yeast Casitone Fatty Acids (YCFA) medium included 4.5 g/L yeast isolate, 3.0 g/L tryptone, 3.0 g/L peptone, 0.4 g/L bile salt, and 0.8 g/L cysteine hydrochloride. The YCFA medium was prepared by dissolving five g/L NaCl, 2.5 g/L KCl, 0.45 g/L MgCl2, 0.2 g/L CaCl2, 0.4 g/L KH2PO4, along with 1.0 mL of Tween 80, 1.0 mL of resazurin, and 2.0 mL of a trace elements solution per 100 mL of the final solution. After thorough dissolution and boiling, 4.5 mL of the prepared medium was transferred into a nitrogen-purged vial, sealed, and sterilized by autoclaving. The gut microbiota extraction solutions from healthy individuals, as well as those with IBS-C and IBS-D, were inoculated into YCFA medium and labeled as the healthy model group, IBS-C model group, and IBS-D model group, respectively. Ginsenosides (0.1 g/100 mL) were added to the YCFA medium to establish ginsenosides-treated groups (H_Z, C_Z, and D_Z) for the healthy, IBS-C, and IBS-D model groups, respectively. Control groups (H_CK, C_CK, and D_CK) were prepared with pure YCFA medium.

16S rRNA gene sequencing

Genomic DNA from the gut microbiota was extracted using the FastDNA® Spin Kit for Soil (MP Biomedicals, Santa Ana, CA, USA) according to the provided instructions. The V3-V4 hypervariable regions of the bacterial 16S rRNA gene were amplified utilizing the GeneAmp 9700 PCR system (ABI, San Diego, CA, USA) with primers 341F (5′-CCTAYGGGRBGCASCAG-3′) and 806R (5′-GGACTACHVGGGTWTCTAAT-3′), according to established protocols (Pi et al., 2022). Following amplification, equimolar amounts of the purified amplicons were pooled and sequenced using the NovaSeq PE250 platform (Illumina, San Diego, CA, USA), according to the standard procedures set by Majorbio Bio-Pharm Technology Co. Ltd. (Shanghai, China). Amplicon sequence variants (ASVs) were identified through DADA2 within the QIIME2 platform (version 2020.2), following performing sequence quality control and denoising. Taxonomic classification of the ASVs was conducted using the Naive Bayes classifier integrated in QIIME2, with reference to the SILVA 16S rRNA database (version 138). All sequence data generated from the in vitro fermentation samples were archived in the National Center for Biotechnology Information (NCBI) Short Read Archive (SRA) under accession number PRJNA1086138.

Bioinformatic analysis

To reduce sampling variability, rare operational taxonomic units (OTUs) with a relative abundance of less than 1% were excluded from subsequent analyses of α- and β-diversity. The Ace index was used to assess α-diversity, while β-diversity was examined through both weighted and unweighted UniFrac-based principal component analysis (PCA) using R software in conjunction with the vegan package (version 3.3.1). Furthermore, the study implemented the linear discriminant analysis effect size (LEfSe) method to ascertain the influence of differentially abundant taxa and to identify those with the most significant biological relevance between the two groups (Segata et al., 2011). Comparisons of the data were carried out using the Wilcoxon rank-sum test and Welch’s t-test, establishing statistical significance at a threshold of P < 0.05.

Statistical analysis

The analysis of data, excluding bioinformatic information, was conducted using SPSS 12.0 software (IBM Corp., Armonk, NY, USA). To compare the results between the ginsenosides treatment groups and the control groups across different model groups, Student’s t-test was employed for qualitative data that met the assumption of equal variance. In cases where the variance was unknown, Welch’s t-test was utilized. A statistical significance threshold was established at P < 0.05.

Results

ASV analysis

Pan analysis revealed a steady rise in the total species count within each group as the sample size expanded (Fig. 1A), indicating a positive correlation between sample size and species diversity. In contrast, core analysis showed that the number of core species in each group plateaued as the sample size increased, suggesting that core microbiota reach a point of stability despite further sample inclusion (Fig. 1B). The rank-abundance curves at the ASV level displayed a smooth, gradual slope, indicating an even distribution of species within the community samples, reflecting the balanced species composition in the analyzed microbiota (Fig. 1C).

Figure 1 ASV analysis.

(A) Pan and (B) core analysis. The horizontal axis represents the number of samples observed, and the vertical axis represents the number of all (Pan)/core species sampled under a given grouping category. (C) ASV rank-abundance distribution curve. The horizontal axis represents the species (ASV) rank; the vertical axis indicates the relative abundance of the species (ASV) in that rank.

Diversity analysis

The α-diversity of the treated groups, receiving ginsenosides (H_Z, C_Z, and D_Z), exhibited no statistically significant alterations when compared to their respective control groups (H_CK, C_CK, and D_CK) using the Ace index in the healthy, IBS-C, and IBS-D models (Figs. 2A–2C). A comparison of the β-diversity of the treated groups with their respective controls (H_Z, C_Z, and D_Z) revealed no statistically significant differences in the healthy, IBS-C, and IBS-D model groups (Figs. 2D–2G).

Figure 2 Diversity of gut microbiota.

α-diversity at the genus level was assessed using the Ace index in healthy (A), IBS-C (B), and IBS-D (C) model groups. β-diversity at the genus level was visualized through PCA (D) and assessed using the box plots in healthy (E), IBS-C (F), and IBS-D (G) model groups. Statistical significance thresholds: P < 0.05.

Microbial composition analysis

The Venn diagram was performed in the healthy model groups (H_CK and H_Z), IBS-C model groups (C_CK and C_Z), and IBS-D model groups (D_CK and D_Z), revealing 51 genera shared among them (Fig. 3A). Community barplot analysis was performed to examine the distribution of different genera (Fig. 3B). To further elucidate these results, pie plots were employed to visually represent the relative abundance of microbial communities at the genus level in different groups (Figs. 3C–3H, Table 1). In the healthy model groups, the most abundant genera were Bifidobacterium (25.21%), Lactobacillus (12.12%), Megasphaera (11.74%), and Escherichia-Shigella (9.52%) in the control group (H_CK), whereas the predominant genera were Bifidobacterium (28.87%), Escherichia-Shigella (19.18%), Megamonas (12.55%), and Lactobacillus (9.20%) in the ginsenosides-treated group (H_Z). Similarly, in the IBS-C model groups, the dominant genera included Bifidobacterium (42.08%), Lactobacillus (17.02%), Megamonas (11.15%), and Escherichia-Shigella (9.51%) in the control group (C_CK), whereas the prevalent genera were Bifidobacterium (45.21%), Lactobacillus (15.58%), Escherichia-Shigella (11.59%), and Megamonas (10.43%) in the ginsenosides-treated group (C_Z). Moving to IBS-D model groups, the dominant genera included Escherichia-Shigella (16.68%), Streptococcus (16.05%), Megamonas (12.11%), and Clostridium_sensu_stricto_1 (10.92%) in the control group (D_CK), while the dominant genera included Lactobacillus (18.45%), Escherichia-Shigella (17.52%), Megamonas (17.25%), and Streptococcus (13.15%) in the ginsenosides-treated group (D_Z).

Figure 3 Composition of gut microbiota.

Venn diagram displaying the numbers of unique and shared microbial communities at the genus level among the experimental groups (A). Community barplot analysis at the genus level visually represents the relative abundance of gut microbiota in the experimental groups (B). Pie plots showing the relative abundance of gut microbiota composition at the genus level in the H_CK(C), H_Z(D), C_CK(E), C_Z(F), D_CK(G), and D_Z(H) groups, respectively.

Table 1 Composition of gut microbiota.

The relative abundance of the predominant genera in different groups.

Model group	Group	Predominant genera	
Healthy model groups	H_CK group	Bifidobacterium (25.21%)	Lactobacillus (12.12%)	Megasphaera (11.74%)	Escherichia-Shigella (9.52%)	
H_Z group	Bifidobacterium (28.87%)	Escherichia-Shigella (19.18%)	Megamonas (12.55%)	Lactobacillus (9.20%)	
IBS-C model groups	C_CK group	Bifidobacterium (42.08%)	Lactobacillus (17.02%)	Megamonas (11.15%)	Escherichia-Shigella (9.51%)	
C_Z group	Bifidobacterium (45.21%)	Lactobacillus (15.58%)	Escherichia-Shigella (11.59%)	Megamonas (10.43%)	
IBS-D model groups	D_CK group	Escherichia-Shigella (16.68%)	Streptococcus (16.05%)	Megamonas (12.11%)	Clostridium_sensu_stricto_1 (10.92%)	
D_Z group	Lactobacillus (18.45%)	Escherichia-Shigella (17.52%)	Megamonas (17.25%)	Streptococcus (13.15%)	

Microbial difference analysis

To evaluate whether particular microbial taxa were differentially enriched among the healthy, IBS-C, and IBS-D model groups, we conducted LEfSe analysis utilizing linear discriminant analysis (LDA). In the healthy model groups (Fig. 4A), six genera displayed significant differences (LDA > 2) between the control group (H_CK) and the ginsenosides-treated group (H_Z). Genera such as UCG-002, Lachnospiraceae_NK4A136_group, Flavonifractor, and UCG-003 were significantly enriched in the H_CK group, while Stenotrophomonas and Achromobacter were significant enriched in the H_Z group. For the IBS-D model groups (Fig. 4B), eight genera demonstrated differential abundance (LDA > 2) between the control group (D_CK) and the ginsenosides-treated group (D_Z). The D_CK group exhibited significant enrichment of Ruminococcus_gnavus_group, Roseburia, Blautia, unclassified_f_Lachnospiraceae, Anaerostipes, and Lachnospira, while Pseudomonas and Stenotrophomonas were predominantly enriched in the D_Z group. Notably, no genera displayed significant enrichment (LDA > 2) were observed between the control group (C_CK) and the ginsenosides-treated group (C_Z) in the IBS-C model groups.

Figure 4 Difference of gut microbiota.

LEfSe analysis at the genus level revealed the specific microbial communities whose abundances were differentially enriched between the control group and the ginsenoside-treated group. in the healthy (A) and the IBS-D (B) model groups. Statistical significance thresholds: LDA > 2.

Discussion

The primary objective of this study was to examine the differential effects of ginsenosides on gut microbiota in healthy individuals compared to patients with IBS. Ginsenosides were incorporated into an in vitro fermentation system using gut microbiota samples derived from both healthy participants and patients diagnosed with IBS-C and IBS-D. A comprehensive analysis through 16S rRNA sequencing provided insights into the resulting changes in gut microbiota composition, highlighting the specific microbial alterations in response to ginsenosides treatment across the different model groups.

In vitro fermentation models are unable to account for a number of critical factors that are present in vivo, including gut absorption, gastrointestinal secretions, and the defense mechanisms of the host. Furthermore, in vivo studies are subject to a number of challenges, including ethical considerations, high costs and lengthy timeframes, which have the potential to complicate the execution of research (Gibson & Fuller, 2000). In our laboratory, we have developed an in vitro simulated gut fermentation system, the efficacy of which has been validated through extensive experimentation, supporting the growth of 60–80% of bacterial species found in the guts of Chinese individuals (Pi et al., 2024). This capability allows us to design targeted experiments that investigate the metabolic interactions between gut microbiota and various components, particularly in the context of digestive disorders. Recently, the human gut simulated fermentation model has gained popularity among researchers as an effective tool for studying the interactions between functional components and the host gut microbiota (Feng et al., 2023; Wu et al., 2023).

By the ASVs analysis, it can be concluded that the observed trends in the pan and core analyses, as well as the species abundance curves, are consistent with a well-sampled and diverse gut microbiota. The stabilization of the pan analysis indicates that the majority of microbial diversity has been captured, while the leveling off of the core analysis suggests the existence of a core set of OTUs common to the sampled population (Liao et al., 2019). The observed decline in species abundance curves is indicative of the presence of a few highly abundant species and a multitude of rare species, a phenomenon that is characteristic of ecological communities (Su et al., 2017).

This study explores the impact of ginsenosides treatment on the diversity of gut microbiota in healthy individuals and those with IBS subtypes. The Ace index was used to assess α-diversity, while PCA was employed to analyze β-diversity. The results indicate that ginsenosides treatment does not significantly affect the diversity of gut microbiota in these populations. Ginsenosides, the active compounds in Panax ginseng, have been the subject of extensive research into their therapeutic effects on a range of health conditions, including their interaction with gut microbiota (Kim et al., 2017; Zhao et al., 2023a). The results of unchanged gut microbiota diversity in this study align with certain previous research, reinforcing the reliability of the results obtained. While ginsenosides may not directly affect the overall diversity of gut microbiota, their therapeutic potential in alleviating symptoms associated with IBS remains significant. This underscores the importance of further exploration into the alternative mechanisms by which ginsenosides may exert their influence, such as by modulating specific microbial taxa or metabolic pathways. The cumulative evidence from related studies suggests that ginsenosides can indeed affect the composition and functionality of the gut microbiota, potentially contributing to their therapeutic effects. Nonetheless, our study reveals that ginsenosides treatment did not lead to any significant alterations in gut microbiota diversity among healthy individuals and those with various IBS subtypes. This lack of significant change could be attributed to several factors, including variations in the demographics of the study populations, the particular types of ginsenosides used, their dosages, and the durations of treatment. A plausible explanation is that ginsenosides exert more pronounced effects on microbioal diversity under pathological conditions or that their impact is subtler, requiring more sensitive analytical methods or extended intervention periods to detect meaningful shifts in diversity.

The Venn diagram analysis revealed the presence of a core set of 51 genera in both healthy individuals and those with IBS subtypes. This indicates the presence of shared microbial taxa across different health states. For example, the prevalence of genera such as Bifidobacterium and Lactobacillus in healthy model groups is consistent with the results of numerous studies that have demonstrated the beneficial roles of these genera in maintaining gut health and overall well-being (Li et al., 2023). Furthermore, the effects of ginsenosides on the composition of the gut microbiota vary depending on whether the subject is healthy or suffers from IBS. In healthy individuals, ginsenosides increased the beneficial bacteria Bifidobacterium and potentially harmful bacteria Escherichia-Shigella, while exhibiting a slight decrease in the beneficial bacteria Lactobacillus. In IBS-C groups, ginsenosides enhanced the population of Bifidobacterium but also increased the population of Escherichia-Shigella, with a reduction in the population of Lactobacillus. In IBS-D groups, ginsenosides were observed to enhance the levels of Lactobacillus, while simultaneously elevating Escherichia-Shigella and reducing Streptococcus. The common genera, such as Bifidobacterium and Lactobacillus, have been demonstrated to play a beneficial role in maintaining gut health (Li et al., 2023). Conversely, Escherichia-Shigella and Streptococcus have been associated with dysbiotic patterns in the gut (Shin et al., 2024; Wang et al., 2023; Zhang et al., 2022). These results suggest that ginsenosides exert a bidirectional influence on the gut microbiota, promoting the growth of both beneficial and harmful bacteria.

Ginsenosides, the active compounds in ginseng, have been demonstrated to regulate the gut microbiota (Zhuang et al., 2021). These compounds have been identified as capable of promoting beneficial bacteria, including Bifidobacterium and Akkermansia, while simultaneously inhibiting pathogenic bacteria (Chen et al., 2022; Zhao et al., 2023a). This suggests a potential therapeutic role in the management of gut health. The results of the LEfSe analysis provide valuable insights into the differential enrichment of specific bacterial types among the healthy, IBS-C, and IBS-D model groups with the ginsenosides treatment. Ginsenosides treatment in the healthy model groups has been shown to alter the composition of the gut microbiota in healthy individuals, with notable changes in the abundance of certain bacterial genera. The changes in gut microbiota composition, including the increase in Stenotrophomonas and Achromobacter and the decrease in UCG-002, Lachnospiraceae_NK4A136_group, Flavonifractor, and UCG-003, suggest that ginsenosides may influence gut barrier function and immune regulation. While the specific studies cited do not directly mention these bacterial genera, the overall findings support the concept that ginsenosides may have a significant impact on the gut microbiota (Dong et al., 2017; Huang et al., 2017; Xie et al., 2022; Zhuang et al., 2021). In the IBS-D model groups, treatment with ginsenosides resulted in a significant change in the composition of gut microbiota compared to the control group. The changes in the gut microbiota associated with IBS-D may be modulated by ginsenosides, as suggested by the enrichment of Pseudomonas and Stenotrophomonas and the depletion of Ruminococcus_gnavus_group, Roseburia, Blautia, unclassified_f_Lachnospiraceae, Anaerostipes, and Lachnospira. This modulation may result in therapeutic effects. Pseudomonas and Stenotrophomonas are genera commonly associated with opportunistic pathogens and dysbiosis (Blanchard & Waters, 2022; Van der Wielen et al., 2023). On the other hand, the genera Ruminococcus, Roseburia, Blautia, Anaerostipes, and Lachnospira are known for their roles in producing butyrate, a short-chain fatty acid (SCFA) that serves as a primary energy source for colonocytes and has anti-inflammatory properties, contributing to gut health and integrity. The depletion of these butyrate-producing bacteria could disrupt gut homeostasis and exacerbate conditions such as IBS-D (Anand, Kaur & Mande, 2016). The lack of significant changes in gut microbiota composition between the control and ginsenosides-treated groups in the IBS-C model groups suggests that ginsenosides may exert their therapeutic effects through mechanisms other than modulation of gut microbiota in IBS-C. This finding could indicate that the dysbiotic profile in IBS-C is less responsive to ginsenosides treatment compared to IBS-D, highlighting the heterogeneity of gut microbiota alterations in different subtypes of IBS. Further research is needed to elucidate the underlying mechanisms and potential therapeutic targets for IBS-C.

While this study offers significant insights, it is essential to recognize its limitations. One limitation of this study is its focus on short-term effects, which may not fully capture the long-term impact of ginsenosides treatment on gut microbiota diversity. Moreover, the use of in vitro fermentation models may not fully replicate the complexities of the gut environment in vivo. Methodological constraints, such as sample size and sequencing depth, could also influence the robustness of the results. In light of these limitations, it is imperative that future research employ larger sample sizes, longitudinal designs, and more advanced analytical techniques to validate the findings.

Conclusions

This study investigated the distinct effects of ginsenosides on gut microbiota composition in healthy individuals and IBS subtypes (IBS-C and IBS-D) through in vitro fermentation. Analysis revealed that ginsenosides selectively altered the abundance of specific microbial taxa without significantly affecting overall microbiota diversity. A core set of 51 genera was conserved across both healthy individuals and IBS patients, underscoring a shared microbial framework. Key genera such as Bifidobacterium and Lactobacillus—critical for gut health—were consistently identified. Notably, ginsenosides differentially modulated gut microbiota composition between healthy and IBS groups, promoting beneficial taxa like Bifidobacterium while increasing potentially pathogenic taxa such as Escherichia-Shigella. This dualistic effect highlights the complexity of ginsenoside activity and their potential therapeutic relevance for IBS symptom management. In IBS-D models, ginsenoside treatment induced marked microbial shifts, characterized by enrichment of Pseudomonas and Stenotrophomonas alongside reduced abundance of butyrate-producing bacteria, which may destabilize gut homeostasis and aggravate symptoms. These findings underscore the multifaceted role of ginsenosides in microbiota regulation, supporting their potential application as IBS therapeutics. Further studies are needed to clarify the mechanistic basis of these effects and their long-term implications for gut health.

The authors utilized ChatGPT (OpenAI) exclusively for language polishing and grammatical refinement during manuscript preparation. The tool was not involved in research design, data analysis, or scientific interpretation.

Additional Information and Declarations

Competing Interests

Author Contributions

Ethics

Data Availability

The authors declare there are no competing interests.

Zhi Du conceived and designed the experiments, authored or reviewed drafts of the article, and approved the final draft.

Chengman Zhao conceived and designed the experiments, authored or reviewed drafts of the article, and approved the final draft.

Jiabin Li performed the experiments, prepared figures and/or tables, and approved the final draft.

Yan Shen performed the experiments, prepared figures and/or tables, and approved the final draft.

Guofei Ren analyzed the data, prepared figures and/or tables, and approved the final draft.

Jieying Ding analyzed the data, prepared figures and/or tables, and approved the final draft.

Jing Peng analyzed the data, prepared figures and/or tables, and approved the final draft.

Xiaoli Ye performed the experiments, prepared figures and/or tables, and approved the final draft.

Jing Miao conceived and designed the experiments, authored or reviewed drafts of the article, and approved the final draft.

The following information was supplied relating to ethical approvals (i.e., approving body and any reference numbers):

The study protocol was reviewed and approved by the Ethics Committee of Hangzhou Centers for Disease Control and Prevention.

The following information was supplied regarding data availability:

The sequence data from the in vitro fermentation samples are available at the Sequence Read Archive (SRA): PRJNA1086138.

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
