# Peer review of "Ginsenosides and gut microbiota: differential effects on healthy individuals and irritable bowel syndrome subtypes"

_PeerJ, doi:10.7717/peerj.19223_

## Round 0.1 · original submission · Minor Revisions

I think that you have clearly outlined the strengths and potential limitations of your study, especially acknowledging that in vitro fermentation is not a perfect way of approaching the study of drug effects on gut bacteria. Clearly some of reviewers were of the same mind, while others were not convinced of the clinical relevance of the findings or the Methodology. Concerns about how exciting the results are not relevant as long as they are sound, and there are no specific concerns expressed about that by any of the reviewers. I am suggesting "minor revisions" simply in order to give you the opportunity to check "Methods" Section of the Ms, and double check for any typographical or expression issues.

Reviewer 1 ·

Basic reporting

English language of the manuscript needs some corrections.

Experimental design

The experimental design is not very robust and has many lacunae. The in vitro fermentation system used in the manuscript cannot replicate in vivo models and therefore the outcome from such a system is doubtful.

Validity of the findings

The outcome of the study is not interesting.

Additional comments

The manuscript entitled “Ginsenosides and gut microbiota: differential effects on healthy individuals and irritable bowel syndrome subtypes” by Du et al. tries to investigate the effects of ginsenosides on the gut microbiota of healthy participants and IBS patients. The manuscript has some major concerns and cannot be considered for publication. The authors have failed to convey what they are looking into and what is the clinical application of this study. It is not clear from the study if the authors want to emphasize the role of ginsenosides as potential therapeutic agents for IBS or if they are just looking into the effects of ginsenosides on the gut microbiota. If they are proposing the use of ginsenosides in IBS therapeutics, then the study is not significant as the authors found no statistically significant microbial alterations in control and ginsenosides-treated groups, suggesting that ginsenosides do not affect the overall diversity of gut microbiota. Moreover, the authors found that ginsenosides also increase harmful bacteria Escherichia-Shigella and decrease butyrate-producing beneficial bacteria. Any treatment that increases harmful gut microbes and decreases butyrate-producing microbes will affect the overall gut health, exacerbating the disease. These findings clearly suggest that ginsenosides may have negative effects on gut microbiota. Terms like ‘pan genomic’ and ‘core genomic’ are used inappropriately in this manuscript. The methodology is not very clear. The major limitations of this study are the use of in vitro fermentation that cannot replicate in vivo model and a small sample size. The results of the findings are also not interesting.

·

Basic reporting

The article conform to professional standards of courtesy and expression; The references fit the broader field of the literature; The structure of this article conform to an acceptable format of ‘standard sections’ ; Figures and raw data are relevant to the content of the article and appropriately described;

Experimental design

Ginsenosides have shown potential in alleviating IBS symptoms, but their interactions with gut microbiota in diferent IBS subtypes are not well-studied. This article investigated the effects of ginsenosides on the gut microbiota of both healthy participants and participants suffering from IBS. The investigation conducted rigorously and in conformity with the prevailing ethical standards in this field.
Methods described with sufficient information.

Validity of the findings

The results elucidate the microbial composition assessmentrevealed the presence of 51 shared genera, with notable variations in composition and a significant enrichment of specific taxa. Specifically, furthermore, the LEfSe analysis revealed that, following ginsenosides treatment, the healthy model groups exhibited significant enrichment of Stenotrophomonas and Achromobacter, while the IBS-D model
groups demonstrated significant enrichment of Pseudomonas and Stenotrophomonas.

The conclusions were appropriately stated and connected to the original question investigated.

Additional comments

no comment

·

Basic reporting

The manuscript is well written.
The study highlighted the necessity for further investigation into targeted microbiome therapies for IBS,
The introduction need expansion, include biologoc therapies for IBS.
Kindely revise the methodology section according to PeerJ journal guidelines.
Please also proof read the whole manuscript.

Experimental design

Well written

Validity of the findings

Novel outcomes

Additional comments

none

Reviewer 4 ·

Basic reporting

no comment

Experimental design

I am concerned about the conclusions of this paper based on the following two points on experiment design: 1. in vitro fermentation may change the composition of the original fecal flora; 2. this administration way of ginsenosides may exerts different influence on fecal flora compared with oral administration.

Validity of the findings

no comment

---

## Round 0.2 · accepted · Accept

Thanks for addressing the concerns of the reviewers around language and explanation of the Methodology. I have no further concerns about the readiness of this study for publication.